# Study of the Expression and Function of Calcium-Sensing Receptor in Human Skeletal Muscle

**DOI:** 10.3390/ijms22147282

**Published:** 2021-07-06

**Authors:** Cecilia Romagnoli, Preeti Sharma, Roberto Zonefrati, Gaia Palmini, Elena Lucattelli, Donald T. Ward, Isabella Ellinger, Marco Innocenti, Maria Luisa Brandi

**Affiliations:** 1Department of Experimental and Clinical Biomedical Sciences, University of Florence, 50139 Florence, Italy; cecilia.romagnoli@unifi.it (C.R.); preeti.sharma@unifi.it (P.S.); roberto.zonefrati@unifi.it (R.Z.); gaia.palmini@unifi.it (G.P.); 2Fondazione Italiana Ricerca sulla Malattie dell’Osso (FIRMO Onlus), 50141 Florence, Italy; 3Plastic and Reconstructive Microsurgery, Careggi University Hospital, 50139 Florence, Italy; elena.lucattelli@gmail.com (E.L.); marco.innocenti@unifi.it (M.I.); 4Faculty of Biology, Medicine and Health, The University of Manchester, Manchester M13 9PL, UK; d.ward@manchester.ac.uk; 5Institute for Pathophysiology and Allergy Research, Medical University of Vienna, 1090 Vienna, Austria; isabella.ellinger@meduniwien.ac.at

**Keywords:** skeletal muscle, satellite cells, myogenesis, calcium-sensing receptor, G protein-coupled receptors

## Abstract

Skeletal muscle has an outstanding capacity for regeneration in response to injuries, but there are disorders in which this process is seriously impaired, such as sarcopenia. Pharmacological treatments to restore muscle trophism are not available, therefore, the identification of suitable therapeutic targets that could be useful for the treatment of skeletal reduced myogenesis is highly desirable. In this in vitro study, we explored the expression and function of the calcium-sensing receptor (CaSR) in human skeletal muscle tissues and their derived satellite cells. The results obtained from analyses with various techniques of gene and protein CaSR expression and of its secondary messengers in response to calcium (Ca^2+^) and CaSR drugs have demonstrated that this receptor is not present in human skeletal muscle tissues, neither in the established satellite cells, nor during in vitro myogenic differentiation. Taken together, our data suggest that, although CaSR is a very important drug target in physiology and pathology, this receptor probably does not have any physiological role in skeletal muscle in normal conditions.

## 1. Introduction

Skeletal muscle tissue has an inherent capacity for regeneration in response to minor injuries. However, there are circumstances that can cause irreversible loss of muscle mass and function, for example, severe trauma, tumor ablation, and different types of congenital or acquired muscle diseases, such as muscular dystrophies and sarcopenia [1,2,3,4,5,6].

Pathological situations, in which skeletal muscle regeneration is seriously impaired, represent conditions that affect motor unit function with significant disabilities and reduced quality of life (QoL) in people who are affected. Pharmacological treatments are not available for these muscle disorders. Hence, there is an unmet need for pharmacological strategies for the treatment of skeletal muscle degenerative disorders.

Postnatal skeletal muscle stem cells, called satellite cells (SCs), are responsible for skeletal muscle regeneration during the lifespan. Several studies report that some myopathies, such as sarcopenia and muscular dystrophies, are related to a failure of SCs to enter into the myogenic program [7,8]. As SCs are committed myogenic precursors, and possess the capability of self-renewal, a unique feature of stem cells, they play a central role in the search for therapies for skeletal muscle disorders [9]. The identification of new therapies, and the testing of candidate drugs for their ability to improve muscle regeneration, require appropriate in vitro cell-based screening assays to select compounds or new targets for further development. The use of human skeletal muscle-derived cells is an essential and extremely powerful cell-based model system that may support and predict from in vivo results the in vitro effects obtained in the laboratory, which could later be useful for finding therapeutic drugs targeted for skeletal muscle degenerative myopathies [10].

In the last decade, it has become possible to isolate and culture a population of SCs [11,12,13,14]. While the use of primary cultures of SCs also has some inherent drawbacks, mainly relating to the fact that they are generally slow growing and can undergo a limited number of divisions [15], SCs represent an important in vitro model to understand skeletal muscle physiology and pathology.

Calcium ion (Ca^2+^) is a ubiquitous intracellular signal that regulates a myriad of cellular processes, as it is one of the most specific and selective messengers in nature. It is involved in multiple signaling cascades critical for cell survival, growth, differentiation, and death [16,17]. It is essential for muscle function, and the tight coupling of muscle excitation and contraction by Ca^2+^ dynamics has been well established. Voltage-gated Ca^2+^ channels are key transducers which are activated by membrane depolarization and mediate Ca^2+^ influx in response to an action potential. Ca^2+^ entering the cell through voltage-gated Ca^2+^ channels serves as the second messenger of electrical signaling that initiates many different cellular events [18]. Moreover, many studies have shown the importance of Ca^2+^ for skeletal muscle development, maintenance, and regeneration [19,20,21].

The calcium-sensing receptor (CaSR) is a key systemic regulator of calcium homeostasis [22], acting as a modulator of extracellular Ca^2+^ concentrations, and it has been proven to be an important molecule in physiology and pathology [16]. It belongs to the class C G protein-coupled receptors (GPCRs), also known as 7-transmembrane domain receptors, a large protein family of receptors that sense molecules outside the cell and activate internal signal transduction pathways and, ultimately, cellular responses [23]. The CaSR is primarily expressed within parathyroid glands where it regulates parathyroid hormone (PTH) secretion, which in turn controls calcium homeostasis, by acting upon kidney, bone, and intestine to maintain Ca^2+^ concentrations within the physiological range (1.1–1.3 mM) [22]. Initially, studies of the CaSR were focused on calciotropic tissues, such as parathyroid, kidney, and bone, which have obvious roles in maintaining calcium homeostasis. Now, it has become clear that the CaSR is expressed in many other cells and organs (such as neurons, lungs, skin, placenta, breast, and blood vessels), without any evident role in maintaining calcium homeostasis [17]. Several studies have observed that abnormal CaSR expression and function are implicated not only in calciotropic disorders, such as hyper- and hypoparathyroidism, but also in diseases linked to non-calciotropic systems, such as of the nervous, reproductive, and respiratory systems, and even in diseases, such as chronic inflammation and cancer [16,17,22].

Drugs targeting CaSR in related pathologies are already available [24]. Moreover, several recent scientific studies have observed a role of CaSR specifically in the physiological differentiation of cells, such as keratinocytes, cardiomyocytes, and neurons, and in smooth muscle contractility and proliferation [25,26,27,28,29].

Despite the importance of this receptor, the expression and function of the CaSR in skeletal muscle has not been explored. Therefore, the aim of our study was to investigate the presence of CaSR in human skeletal muscle and its possible role in skeletal muscle myogenic differentiation, in order to find new possible therapeutic targets for skeletal muscle degenerative diseases.

## 2. Results

### 2.1. Isolation of Primary Culture of Human Satellite Cells (hSCs) and Characterization of Established hSC Lines

Satellite cells (hSCs) were isolated from human skeletal muscle biopsies, as described recently [14], and subcultured in Matrigel-coated plates (Figure 1A,B).

Gene expression of *Pax7*, a nuclear transcriptional factor and the main marker of SCs [30], was confirmed by RT-PCR (Figure 1C). By flow cytometry, Pax7 protein expression was demonstrated in 98.29% of all cells (Figure 1D) and confirmed by Western blot analysis (Figure 1E).

### 2.2. In Vitro Myogenic Differentiation of hSCs

Confluent hSCs, at 70–80% density in the plate, were differentiated toward the myogenic phenotype using an appropriate myogenic differentiation medium for 9 days. During this period, the activated cells align (myoblasts), then subsequently fuse with each other and, finally, differentiate into multinucleated myofibers. Phase contrast microscopy revealed the presence of multinucleated elongated cells, containing from three to more than eight nuclei, related to myotubes (Figure 2A).

Myogenic differentiation was confirmed by expression of the myosin heavy chain (MHC), which is the most important terminal myogenic differentiation marker protein [31]. After 9 days of myogenic induction, the multinucleated cells exhibited high MHC expression, as confirmed by immunofluorescence microscopy (Figure 2B,C).

The myogenic phenotype was also confirmed by RT-PCR analysis to evaluate the expression of the myogenic regulatory factor genes *Myf5, MyoD1, Myogenin*, and the terminal differentiation marker *MHC*. We observed that these genes were expressed at the same time during the last period (9 days) of myogenic differentiation when myotubes were formed, while during the first phase (T_0_) of the myogenic differentiation, only *Myf5* and *MyoD1* were expressed, in accordance with the literature, while *Myogenin* and *MHC* were completely absent (Table 1). The earlier detection of *MyoD1* that we found in the primary cultured hSCs is due to the fact that after collecting biopsies, the surgical procedure per se is a form of injury that initiates muscle regeneration, which leads to the activation of satellite cells and to their commitment to myogenic differentiation.

MHC protein expression was also evaluated by Western blot analysis at different time points (T_0_–3–6–9 days) of myogenic differentiation and revealed a gradual increase in MHC over time during myogenic differentiation progression (Figure 2D), validating the suitability of the in vitro myogenesis model.

### 2.3. Analysis of CaSR Gene Expression in Human Skeletal Muscle Tissues (hSMts) and in the Derived Human Satellite Cell (hSC) Lines and during In Vitro Myogenesis

We performed qualitative RT-PCR using different pairs of primers that target exons 2/3, 4/5, and 6/7 of the human *CaSR* gene (Table 2). CaSR mRNA expression was neither detected in human skeletal muscle tissues (hSMt1, hSMt2, hSMt3), nor in the respective established human satellite cell lines (hSCs1, hSCs2, hSCs3), using any of the primer settings (Figure 3A–C). A primary cell culture of human adenoma parathyroid cells (PTcs) was used as a positive control. On the other hand, human embryonic HEK293, used as a negative control, did not show *CaSR* expression, as expected [32]. Housekeeping gene *β-actin* was used to verify the appropriate quality of the assayed samples (Figure 3D).

Real-time qPCR detected very low *CaSR* expression levels in hSMts and in the established hSC lines, but since the quantity was extremely small in all of the assayed samples, it was not possible to sequence the amplicons to confirm the detected mRNA products (Figure 3E).

Subsequently, since during in vitro myogenesis satellite cells mature and gradually express receptors [14], we analyzed *CaSR* gene expression by RT-PCR at T_0_ and after 9 days of myogenic differentiation, using all three different pairs of primers previously utilized. The analyses showed no *CaSR* gene expression at these experimental time points, demonstrating the absence of *CaSR* gene during the in vitro myogenesis of the three primary hSC lines. *CaSR* gene expression was demonstrated in PTcs, while it was absent from HEK293 cells. Housekeeping gene *β-actin* was used in order to verify the appropriate quality of the assayed samples (Figure 4A–D). Additionally, real-time qPCR was also performed to evaluate the possible expression of *CaSR* during myogenesis. We evaluated *CaSR* gene expression at T_0_, 3, 6, and 9 days of myogenic differentiation of the three primary hSC lines. Data verify the absence of *CaSR* during myogenic induction, confirming the data reported in Figure 3, and assuming that this gene is not present during the differentiation process of the hSCs (Figure 4E,F).

### 2.4. CaSR Protein Expression Analysis in Human Skeletal Muscle Tissues (hSMts), in hSCs, and during In Vitro Myogenesis

After analyzing the *CaSR* gene expression, our study was focused on the evaluation of a possible expression of CaSR protein in hSMt sections, in the established hSC lines, and during myogenic differentiation. We performed: (a) Western blot analysis, (b) immunofluorescence staining, and (c) flow cytometry, in order to assess the presence of the protein CaSR.

CaSR protein expression in primary hSC lines was investigated by Western blot analysis, by which it is possible to evaluate the size of the CaSR protein either in monomers of 140 and 160 kDa (in reducing condition with β-mercaptoethanol or 50 mM DTT, dithiothreitol, disulfide bond breaker) or in dimers of 240 kDa (non-reducing condition) forms.

The analysis, in standard reducing conditions, was performed in primary hSC lines at different time points: T_0_, 3, 6, and 9 days of myogenic differentiation. The specificity of the antibody was demonstrated using HEK293 cells, stably transfected with human CaSR (HEK_CaSR_), and used as a positive control. The obtained results showed very faint bands of about 140 kDa and 160 kDa (size of CaSR monomers) in the basal hSC lines and during myogenic differentiation (Figure 5A).

Since we had not observed the CaSR gene expression in the established hSC lines and during myogenic differentiation, but did observe faint bands in Western blot analyses with similar molecular weight of CaSR monomer bands in the same samples, we analyzed the samples under reducing and non-reducing conditions by Western blot.

The obtained results showed that the monomer CaSR bands, observed in the positive control HEK_CaSR_ in a standard reducing condition, shifted to the CaSR dimer band at 240 kDa in non-reducing conditions. On the other hand, the monomer bands in the hSC lines, obtained in standard reducing conditions, did not shift to the dimer band at 240 kDa in the non-reducing condition (Figure 5B).

Then, immunofluorescence analysis of CaSR protein was performed. The hSMt sections (Figure 6A–C) show how human skeletal muscle appears in phase contrast microscopy, with the longitudinal view of the skeletal muscle fibers.

The results of immunofluorescence analysis revealed no staining of CaSR in hSMt sections (Figure 6D,F). In comparison to that, positive detection of CaSR expression in human parathyroid tissue sections validated our method of analysis (Figure 6G,I). The specificity of the anti-CaSR antibody was verified by one extra internal negative control in the assay, where the antibody binding was blocked by pre-absorption with the blocking peptide [33,34]. The results demonstrate the absence of CaSR protein expression in hSMts (Figure 6E,H).

The subsequent immunostaining performed in hSC lines also showed no detection of CaSR protein in satellite cells (Figure 7A,B). Human PTcs were used as a positive control, to validate the method and anti-CaSR antibody (Figure 7C,D).

In addition to immunostaining, we performed flow cytometry using two different dilutions (1:500 and 1:100) of the primary anti-CaSR antibody (5C10, ADD). The obtained data showed the absence of CaSR protein in hSC lines even at the higher concentration of the anti-CaSR antibody; indeed, with both antibody dilutions used, 1:500 and 1:100, the rate of stained cells is close to zero (0.27% and 0.41%, respectively), confirming the results obtained with immunofluorescence.

### 2.5. CaSR Secondary Messenger Analysis in hSC Lines

CaSR signals to downstream pathways via three main groups of the heterodimeric G protein: Gα_q/11_, Gα_i/0_, and Gα_12/13_ [35]. We verified the absence of CaSR protein expression and function in hSCs by analyzing some secondary messengers upon treating the hSC lines with Ca^2+^ and CaSR drugs: NPS2143 (calcilytic) and NPS R-568 (calcimimetic).

#### 2.5.1. Intracellular Calcium Mobilization Imaging

We performed the analysis of the intracellular calcium mobilization triggered by extracellular calcium in three live hSC lines. The obtained results showed that hSCs do not respond to CaSR stimulation by 0.5 mM, 3 mM, and 5 mM Ca^2+^ and by 1 µM CaSR drugs NPS2143 or 1 µM NPS R-568 along with Ca^2+^, in an intracellular calcium mobilization assay evaluated by Fura 2-AM (Figure 8B), confirming the absence of CaSR protein in the cellular model. HEK_CaSR_, used as a positive control in the assay, responded well to all of the CaSR stimulations (Figure 8A).

#### 2.5.2. CaSR Secondary Messenger Analysis in hSC Lines: Gα_q/11_ and Inositol Triphosphate (IP_3_)

To verify the absence of CaSR protein in hSC lines, we performed the analysis of IP_3_ secondary messenger using an IP_1_ ELISA assay upon inducing hSC lines, independently, in technical quadruplicates with different concentrations of Ca^2+^ (1.2 mM and 3.5 mM) for 1 h. The results are shown in Figure 8C and confirm absence of any Ca^2+^ dose-dependent increase in IP_1_ response in hSCs.

#### 2.5.3. CaSR Secondary Messenger Analysis in hSC Lines: Gα_i/0_ and Cyclic Adenosine Monophosphate (cAMP)

We analyzed the second messenger cAMP upon stimulation with 3 mM Ca^2+^ alone and with CaSR drugs (1 µM NPS R-568 or 1 µM NPS2143), by ELISA. The results show significant decreases in cAMP production in hSC lines with slight effects with 3.0 mM Ca^2+^ treatment (−24%) and with co-treatments with 1 µM NPS R-568 (−32.7%) or 1 µM NPS2143 (−12.7%), with respect to the control with FSK, used to induce the maximum cAMP response in cells (Figure 8D).

### 2.6. Analysis of a CaSR Homologous Protein in Human Skeletal Muscle: GPRC6A

While we confirmed the absence of the CaSR gene and protein expression in hSMts and hSCs, we got a response, albeit low, in the cAMP secondary messenger production upon stimulation of the hSCs with 3.0 mM Ca^2+^ and in co-treatment of CaSR drugs. This evidence led us to consider the presence of a recently discovered CaSR homologous protein, the G protein-coupled receptor class C group 6 member A (GPRC6A). It is reported to respond to calcium with low affinity, and also to the drugs NPS2143 and NPS R-568, and was found to produce cAMP secondary messenger in response to its activation, although these studies used a murine model of GPRC6A [36,37,38]. Considering all the information reported about GPRC6A, we hypothesized the presence of a GPRC6A receptor in human skeletal muscle as a potential drug target for skeletal muscle disorder. To analyze the expression of the GPRC6A gene, we performed RT-PCR in the hSMts and in the established hSC lines, using a commercial human kidney cDNA (hKcDNA) as a positive control. However, the results verified the absence of GPRC6A gene in hSMts and in the established hSC lines (Figure 9).

## 3. Discussion

The study of regenerative mechanisms that regulate skeletal muscle physiology to find new therapeutic drug targets represents a challenging field of research. Satellite cells are the postnatal stem cells of skeletal muscles which, in response to injury or daily wear and tear, undergo myogenesis, therefore being responsible for skeletal muscle regeneration [39]. The failure of satellite cells to undergo a myogenic differentiation program is one of the main causes of skeletal muscle disorders, such as sarcopenia and muscular dystrophies [5,6,40,41]. However, the pathogenetic mechanisms in these myopathies are not well understood.

Thanks to the research advances in the last decade, it has become possible to isolate and culture populations of human satellite cells that have unambiguously shown the capacity to differentiate into myotubes in vitro [11,12,13], providing invaluable tools with which muscle physiology can be studied.

Since Ca^2+^ plays an important role in skeletal muscle development [19], we have hypothesized that CaSR, along with Ca^2+^ channels, can have a role in myogenic differentiation, becoming a possible molecular target for skeletal muscle disorders.

Expression of CaSR in cells with functions unrelated to systemic calcium homeostasis has been demonstrated in many cases, such as in neurons, skin, and placenta [17]. Unlike the case with other muscle tissues like cardiac and vascular smooth muscle cells, there has been little attempt at analyzing GPCRs in skeletal muscle [42]. In particular, CaSR is actually expressed in cardiomyocytes [43] and in airway smooth muscle cells [29]. Therefore, the assumption that this receptor might be present in skeletal muscle is plausible.

The aim of our work was to analyze the CaSR expression in hSMts, in their respective established hSC lines, and during the in vitro myogenic differentiation. In our study, we have analyzed both gene and protein expression, using different techniques and appropriate positive/negative internal controls, in order to strengthen the obtained results. As far as we know, this is the first report that takes into consideration the question of CaSR expression in skeletal muscle.

The presented results did not reveal any evidence of the expression of CaSR in skeletal muscle cells. This conclusion is based on followed lines of proof: (i) transcripts of *CaSR* were not detected in hSMts, in their derived SCs, or during myogenesis using three different pairs of primers which fall in different exons. Moreover, the analysis with a highly sensitive real-time qPCR technique with a TaqMan probe showed very low detectable limit values, in which it was not possible to sequence the amplicons to confirm the presence of mRNA of *CaSR*; (ii) protein expression of CaSR by immunofluorescence staining did not evidence the protein in skeletal muscle tissue or satellite cells, neither by flow cytometry analysis nor by Western blot analysis.

As CaSR is a GPCR, it can activate many G proteins, providing biologically important amplification steps in the signaling pathway. This underlines the importance of even a very low level of expression in physiological and pathological conditions [44]. Thus, to evaluate the possibility that the presence of *CaSR* mRNA and protein might be very low in skeletal muscle tissues and in the established cellular model of satellite cells, and therefore the inability to detect by *CaSR* probe and anti-CaSR antibodies used in our analysis, we performed analyses of CaSR functionality by studying CaSR secondary messengers (intracellular calcium mobilization triggered by extracellular calcium, IP_3_, and cAMP), using the main ligand of CaSR Ca^2+^, calcimimetic NPS R-568 (positive CaSR allosteric modulator), and calcilytic NPS2143 (negative CaSR allosteric modulator) [24].

The analysis of the intracellular calcium mobilization triggered by stimulation with increasing calcium concentrations, alone and together with calcilytic NPS2143 or calcimimetic NPS R-568, showed no detection of signal variations in intracellular calcium, supporting the absence of CaSR in hSCs, and as we expected and found in HEK_CaSR_, used as a positive control for our experiments. If our hypothesis was true, and if skeletal muscle had the CaSR, our data would be in contrast with results in the literature which report that an increase in extracellular calcium induced an increment in intracellular calcium concentration in cells, such as cardiac myocytes, where the receptor is functionally present [35,43].

The analysis of CaSR secondary messenger IP_3_ in the Gα_q_/11 CaSR signaling pathway in human satellite cells stimulated with increased calcium concentrations showed no significant results and the absence of a dose response in IP_3_ production. The data are in contrast with previous reports in cardiomyocytes and in pulmonary arterial smooth muscle cells where CaSR is functionally expressed, and calcium and calcimimetic induce IP_3_ accumulation [35,43,45]. However, a study reported that in aortic vascular smooth muscle cells, stimulation with calcium did not causes an IP_3_ increase, although the authors detected the expression of CaSR mRNA and protein [46].

In order to have an overall analysis of CaSR functionality in hSCs, cAMP secondary messenger in the Gα_i/0_ CaSR signaling pathway has been investigated, because CaSR signaling pathways can vary from cell to cell, depending on their specific physiology [17]. In our cellular model, we detected significant variations in cAMP response in hSCs stimulated with 3 mM Ca^2+^ alone and together with calcilytic or calcimimetic, in agreement with a study on juxtaglomerular cells expressing CaSR, that if treated with the CaSR agonist cinacalcet, showed a significant decrease in cAMP [47].

Since data on cAMP were the only data supporting a role for CaSR in skeletal muscle cells, a possible explanation is that a homologous receptor related to CaSR is present, as suggested for osteoblasts and osteoclasts [48,49,50].

A CaSR homologous protein is the GPRC6A receptor, a recently discovered physiologically important GPCR [36,51], which responds to Ca^2+^ and to calcilytic (NPS 2143) and calcimimetic (NPS R-568) through a cAMP secondary messenger, although studies have been performed using mouse models [36,37,38]. Considering this information, we hypothesized the presence of GPRC6A, instead of CaSR, in our model, which could be a possible drug target for skeletal muscle disorder. However, although a study in the literature reports the expression of the GPRC6A gene from multiple tissue cDNA panels that contain normalized adult human cDNA preparations, including skeletal muscle [51], the analysis of GPRC6A gene expression, performed in hSMts and in cultured hSC lines, gave negative data. The mechanisms responsible for the observed changes in cAMP levels remain unknown and need to be analyzed in the future.

## 4. Materials and Methods

### 4.1. Primary Culture and Development of Lines of Human Satellite Cells (hSCs)

Human skeletal muscle biopsies were obtained from the discarded tissues of patients undergoing plastic surgery, after signing an informed consent in accordance with a protocol approved by the Local Ethics Committee of the University Hospital Careggi (AOUC), Florence (Italy), for human studies (Rif. N. 16.022), as well as the ethical standards stated in the Declaration of Helsinki (1964) and its later amendments or comparable ethical standards. The biopsies were put in Dulbecco’s modified Eagle medium (DMEM) and, within 24 h of collection, they were processed in the laboratory to obtain primary cultures, called hSCs [14]. Cells at the early passages (from 1 to 4) were used for all experiments.

### 4.2. In Vitro Myogenic Differentiation

In order to differentiate hSCs, cells were cultured to 60–70% confluency in Matrigel (BD Company, Franklin Lakes, NJ, USA) -coated culture dishes and subsequently induced with myogenic differentiation medium (MDM) for 9 days, as previously reported [14].

### 4.3. Flow Cytometry Analysis

Flow cytometry was performed to characterize the cultured hSC lines for the presence of Pax7 protein, using anti-Pax7 antibody (Sigma Aldrich, St. Louis, MO, USA, SAB1412356) dilution 1:10, and the presence of CaSR protein using anti-CaSR (5C10, ADD) antibody (Thermo Fisher Scientific, Waltham, MA, USA, MA1-934) at two different dilutions of 1:500 and 1:100, both followed by goat anti-mouse IgG (H + L) Superclonal™ Secondary Antibody, Alexa Fluor^®^ 488 conjugate (Thermo Fisher Scientific, Waltham, MA, USA, A28175, dilution 1:200). The FIX & PERM^®^ Cell Permeabilization Kit (Invitrogen, Walthan, MA, USA, GAS004) was used for fixing and permeabilizing cells in suspension. Then, 1 × 10^5^ cells were labeled with primary antibodies for 20 min at room temperature (RT) in PBS with 1% bovine serum albumin (BSA), then removed and washed two times and centrifuged for 5 min at 1500 rpm; secondary antibody was incubated for 30 min at RT in the dark, then washed once and promptly analyzed. The stained cells were analyzed in a CyFlow^®^Space cytometer, equipped with FlowMax^®^ software (Sysmex Partec, Norderstedt, Germany).

### 4.4. Immunofluorescence Staining of Cells and Tissues

Immunofluorescence staining was performed in proliferating hSCs, and after myogenic differentiation, to evaluate the presence of one of the main terminal myogenic differentiation marker MHC proteins, using anti-MHC antibody, clone A4.1025 (Sigma-Aldrich, St.Louis, MO, USA, 05-716) dilution 1:20, and the presence of CaSR protein, using anti-CaSR (5C10, ADD) antibody (Thermo Fisher Scientific, Waltham, MA, USA), dilution 1:500. After washing two times with PBS solution with 5% BSA, goat anti-mouse IgG (H+L) Superclonal™ Secondary Antibody, Alexa Fluor^®^ 488 conjugate (Thermo Fisher Scientific, Waltham, MA, USA, A28175, dilution 1:200) was added for 1 h at RT. Samples were then washed with PBS for observation with laser scanning confocal microscopy (LSCM), using an LSM 510 Meta microscope (ZEISS, Oberkochen, Germany).

Tissue sections (4 µm) were de-waxed and rehydrated. Antigen retrieval was carried out with 0.05% (*v*/*v*) citraconic anhydride solution, pH 7.4, for 20 min. Sections were incubated with blocking buffer (5% (*v*/*v*) goat serum (Jackson ImmunoResearch Laboratories, Ely, Cambridgeshire, UK; 005-000-121) in PBS containing 0.05% (*w*/*v*) saponin (Sigma Aldrich, St.Louis, MO, USA; SAE0073) for 1 h at RT. Anti-CaSR (5C10, ADD) antibody, alone or together with 5X blocking peptide (ADDDYGRPGIEKFREEAEERDI (Thermo Fisher Scientific, Waltham, MA, USA, project: BC101281.1, solubilized in H_2_O) and corresponding secondary antibody goat anti-mouse IgG Alexa Fluor 647 (Abcam, Cambridge, UK, ab150115, dilution 1:1000), diluted in blocking buffer, were applied overnight at 4 °C or for 2 h at room temperature, respectively. In negative control incubations, primary antibody was omitted.

Nuclei were stained with 4′,6-diamidino-2-phenylindole, dihydrochloride (DAPI; Roche Diagnostics GmbH, Basilea, Swiss; 10236276001; 50 µg/mL in PBS). After each incubation step, sections were washed intensively with PBS. Fluoromount-G (SouthernBiotech, Birmingham, AL, USA; 0100-01) was used as mounting medium. Images were acquired using an automated widefield fluorescence microscope (Axio Imager Z1, Zeiss, Oberkochen, Germany), equipped with an EC Plan-Neofluar 20x/0.5 objective (Plan-Neofluar, Zeiss, Oberkochen, Germany ) and the following filter sets (Chroma Technology Corp. Bellows Falls, VT, USA): 49000 ET-DAPI and 49008 ET-mCherry, TxRed in combination with TissueFAXS Image Acquisition and Management Software (Version 6.0; TissueGnostics GmbH, Wien, Austria). Using a monochrome camera (Hamamatsu, Hanamatsu City, Shizuoka, Japan), grayscale images of individual fluorescence channels were acquired. Pseudo-colors were assigned to the individual images and selected fluorescence channels were combined and exported.

### 4.5. Gene Expression Analysis by Qualitative Reverse Transcriptase Polymerase Chain Reaction

Qualitative RT-PCR was performed to characterize the hSC lines and the in vitro myogenic differentiation by analyzing the presence of specific marker genes (*Pax-7*, *Myf5, MyoD1, Myogenin*, and *MHC*). The RT-PCR was also used to evaluate the expression of the *CaSR* gene in human skeletal muscle tissue, in the established hSC lines and during in vitro myogenesis, using different pairs of primers targeting various exons, 2/3, 4/5, and 6/7, of the human CaSR [44]. Finally, the RT-PCR was performed to analyze the expression of the *GPRC6A* gene in human skeletal muscle tissue and in established hSC lines using human GPRC6A-specific primers. The information about primers used in the analyses is given in Table 2.

RNA was isolated from the cellular pellets using QIAzol Lysis Reagent (Qiagen, Hilden, Germany, 79306), according to the manufacturer’s protocol, and 500 ng of the isolated RNA was reverse transcribed using the QuantiTect Reverse Transcription kit (Qiagen, Hilden, Germany, 205310), according to the manufacturer’s protocol. In CaSR gene expression analysis, the cDNA isolated from primary cultured human parathyroid cells (PTcs) and HEK293 cells was used as the positive and negative controls, respectively. Human PTcs were obtained from fresh parathyroid adenoma tissue, as previously reported [52]. HEK 293 (Sigma-Aldrich, St.Louis, MO, USA, 85120602) cells have been reported not to express the CaSR gene and protein, thus they can be used as a good negative control for CaSR expression analysis [32]. The *β-actin* house-keeping gene was used as an internal control. All the obtained amplicons were verified by sequencing analysis using a BigDye™ Terminator v1.1 Cycle Sequencing Kit based on Sanger sequencing reactions (Thermo Fisher Scientific, Waltham, MA, USA, 4337451), according to the manufacturer’s protocol.

### 4.6. Gene Expression Analysis by Real-Time Quantitative Reverse Transcription Polymerase Chain Reaction

Real-time quantitative reverse transcription polymerase chain reaction (real-time qPCR) was performed in human skeletal muscle tissues, in proliferating hSCs, and during myogenic differentiation. The gene included in the analysis was *CaSR*. Target gene expression was normalized to ribosomal protein S18 (*RPS18*). The information of the probe and primers used in the analysis is given in Table 3.

### 4.7. Protein Expression Analysis by Western Blot

Equal amounts of protein samples were first diluted (4:1) with 5× Laemmli buffer, followed by resolution of the proteins by SDS-PAGE and electrophoretic transfer into a nitrocellulose membrane (GE Healthcare Life Sciences, Little Chalfont, Buckinghamshire, UK). Membranes were incubated in blocking buffer solution containing Tween-TBS solution and 5% (*w*/*v*) bovine serum albumin (BSA) for 1 h and probed with primary antibodies for Pax7 (mouse monoclonal primary antibody, dilution 1:3000, Sigma-Aldrich, St. Louis, MO, USA, SAB1412356), MHC (mouse monoclonal primary antibody clone A4.1025, dilution 1:3000, Sigma-Aldrich, St. Louis, MO, USA, 05-716), CaSR (monoclonal mouse antibody 5C10 ADD, dilution 1:3000, Thermo Fisher Scientific, Waltham, MA, USA, MA1-934), or β-actin (β-actin peroxidase monoclonal (AC-15) mouse antibody, dilution 1:25,000, from MERCK, Darmstadt, Germany, A3854) for 1 h. After washing the membranes three times with Tween-TBS solution and 5% BSA, secondary antibody was incubated for 1 h at RT with horseradish peroxidase-conjugated horse anti-mouse IgG (1:10,000) from Cell Signaling, Danvers, MA, USA (Cat. number 7076). The blot was then developed with the ECL kit (Thermo Fisher Scientific, Waltham, MA, USA, 32209) according to the manufacturer’s instructions. The chemiluminescence generated was detected by using a Chemidoc gel analyzer (BIO-RAD, Hercules, CA, USA). The blotting membrane was stored in TTBS buffer at 4 °C for re-probing if required.

HEK_CaSR_ were used as a positive control in the CaSR protein expression analysis [53].

### 4.8. Secondary Messenger Analysis

The secondary messengers Gq-coupled inositol monophosphate (IP_1_), intracellular calcium mobilization, and Gi-coupled cyclic adenosine monophosphate (cAMP) were analyzed in hSC lines to investigate the effects of Ca^2+^ and CaSR drugs, calcimimetic (NPS R568, TOCRIS, Minneapolis, MN, USA, 3815), and calcilytic (NPS 2143, TOCRIS, Minneapolis, MN, USA, 3626), in order to verify the expression analysis of CaSR in this cellular model.

#### 4.8.1. Inositol Monophosphate (IP_1_) ELISA Assay

An IP_1_ ELISA assay kit (Cisbio, Waltham, MA, USA, 72IP1PEA/D rev03) was used to analyze IP_1_ response in hSC lines upon stimulation with different Ca^2+^ concentrations (CaSR ligand). The experiment was performed in 3 primary hSC lines.

For this analysis, the hSCs were cultured in Matrigel (BD Company, Franklin Lakes, NJ, USA) -coated 24 multiwell plates in growth medium. Upon reaching 60–70% confluency of the cells in wells of 24 multiwell plates, the cells were washed once with DPBS and then stimulated independently in quadruplicates of each stimulation with 250 µL of 0.5 mM Ca^2+^, 250 µL of 1.2 mM Ca^2+^, and 250 µL of 3.5 mM Ca^2+^ prepared in experimental buffer (0.5 mM CaCl_2_, 125 mM NaCl, 4 mM KCl, 0.5 mM MgCl_2_, 20 mM HEPES, 0.1% (*w*/*v*) glucose) for 1 h at 37 °C in modified air with 5% CO_2_. Each buffer was freshly prepared and supplemented with 50 mM LiCl; then, pH was adjusted to 7.4 using NaOH and HCl. The IP_3_ lifetime within the cell is very short (less than 30 s) before it is transformed into IP_2_ and IP_1_. When LiCl is added to the stimulation buffers, the degradation of IP_1_ into myo-inositol is inhibited. Therefore, IP_1_ can be accumulated in the cells and be measured. Following incubation, 250 µL of absolute EtOH (frozen at −80 °C) was added to each well and subsequently incubated overnight at −20 °C. The cold temperature of the EtOH stops the reaction and solubilizes the cell membrane in order to remove IP_1_ in buffer solution. The next day, the cell lysate from each well was transferred to 1.5 mL centrifuge tubes. The pipetting was carried out slowly, in order to obtain all the volume and avoid the aspiration of precipitated cell debris and proteins on the surface of the wells. The solution in each vial was lyophilized and dissolved in 50 µL of diluent provided with the IP_1_ ELISA assay kit (Cisbio, Waltham, MA, USA, 72IP1PEA/D rev03). The lyophilization step was performed to concentrate the low IP_1_ response which could be present in the cells. Then, the IP_1_ ELISA assay was performed according to the kit instructions. A one-way ANOVA with Bonferroni’s post hoc test was performed on data obtained to analyze the significant differences in IP_1_ response in the presence of the different concentrations of Ca^2+^.

#### 4.8.2. Intracellular Calcium Mobilization Imaging

The intracellular calcium mobilization imaging analysis was performed in 3 live primary hSC lines, upon stimulation with various Ca^2+^ concentrations and CaSR drugs: calcilytic (NPS 2143) and calcimimetic (NPS R-568), along with Ca^2+^.

For this analysis, the primary hSCs were cultured in growth medium on 13 mm Matrigel-coated glass coverslips for 48 h. For use as a positive control in the analysis, HEK_CaSR_ (HEK239 transfected with human CaSR) cells were cultured on 13 mm glass coverslips in DMEM with 10% FBS for 48 h. The experimental buffer (0.5 mM CaCl_2_, 125 mM NaCl, 4 mM KCl, 0.5 mM MgCl_2_, 20 mM HEPES, 0.1% (*w*/*v*) glucose) was freshly prepared, and pH was adjusted to 7.4 using NaOH and HCl. The cells were loaded with 1 µM FURA2-AM in 1.2 mM Ca^2+^ experimental buffer supplemented with 0.1% (*w*/*v*) BSA for 1-2 h at RT in the dark. Then, cells on a coverslip were mounted in a perfusion chamber and observed through a 40X oil immersion lens on a Nikon Diaphot inverted microscope (Miato, Tokio, Japan) equipped with MetaFluor^®^ Fluorescence Ratio Imaging Software (Molecular Devices, San Jose, CA, USA); dual-excitation-wavelength microfluorometry was then performed. The baseline free ionized Ca^2+^ in cells was measured in 0.5 mM Ca^2+^ experimental buffer. Subsequently, the cells were stimulated serially for 3 min each with 3 mM Ca^2+^, 3 mM Ca^2+^ + 1 µM R568 (calcimimetic), 0.5 mM Ca^2+^, 5 mM Ca^2+^, 5 mM Ca^2+^ + 1 µM NPS2143 (calcilytic), 0.5 mM Ca^2+^ prepared in experimental buffer, and free ionized Ca^2+^ in cells was measured. Each stimulation was performed in technical triplicates.

#### 4.8.3. Cyclic Adenosine Monophosphate (cAMP) ELISA Assay

The secondary messenger cAMP was analyzed using the cAMP complete ELISA kit (Abcam, Cambridge, UK, ab133051) to evaluate the functionality of Ca^2+^/CaSR in primary hSC lines upon stimulation with Ca^2+^ and calcilytic (NPS R-568) and calcimimetic (NPS2143) drugs. The experiment was performed in 3 primary hSC lines.

For the assay, the hSCs were cultured in growth medium in Matrigel (BD Company, Franklin Lakes, NJ, USA) -coated 24-well plates. Upon reaching 60–70% confluency, the cells were washed once in DPBS and then stimulated. Since Ca^2+^/CaSR inhibit the cAMP response in healthy cells [54], the cells for this assay were stimulated with experimental buffers supplemented with forskolin (FSK). FSK is a potent activator of stimulatory G protein (Gs), and stimulates the maximum level of cAMP in the cells. Therefore, FSK was used as an internal positive control for the assay. For the Ca^2+^/CaSR cAMP assay, the primary hSCs were stimulated independently in triplicate for each stimulation in wells of a 24-well tissue dish, with 250 µL of experimental buffers containing 0.5 mM Ca^2+^ (control), 0.5 mM Ca^2+^ + 10 µM FSK (positive control), 3 mM Ca^2+^ + 10 µM FSK (Ca^2+^ effect), 3 mM Ca^2+^ + 10 µM FSK + 1 µM R568 (calcimimetic effect), and 3 mM Ca^2+^ + 10 µM FSK + 1 µM NPS2143 (calcilytic effect), for 15 min at 37 °C in modified air with 5% CO_2_.

Following incubation, 150 µL of HCl 0.1 N + 0.5% Tryton X100 was added to each well and after 15 min at room temperature, the lysates were collected and centrifuged at 1000× *g* to pellet the cellular debris. Then, the cAMP ELISA assay was performed according to the kit instructions (Abcam, Cambridge, UK, ab133051). A one-way ANOVA with Bonferroni’s post hoc test for Ca^2+^/CaSR cAMP was performed on results obtained to analyze the statistical differences.

## 5. Conclusions

In this work, we have presented for the first time the lack of CaSR expression and functionality in human skeletal muscle cells. Although CaSR is a very important receptor in the physiology of several organs and tissues, and also in several pathologies, it is absent from human skeletal muscle tissue, derived SCs, and cells differentiating into myoblasts, proving that CaSR is not involved in myogenesis. As Ca^2+^ is very important in skeletal muscle physiology, it is through voltage-gated Ca^2+^ channels that Ca^2+^ ions enter the cells, functioning as second messengers in the control of cell functions.

Since this is a first study exploring CaSR in healthy human skeletal muscle, future studies should address the same question in the evaluation of CaSR expression in tissues derived from patients with myopathies or other muscular disorders. CaSR could indeed represent an important pharmacological target in pathological conditions.

## Figures and Tables

**Figure 1 ijms-22-07282-f001:**
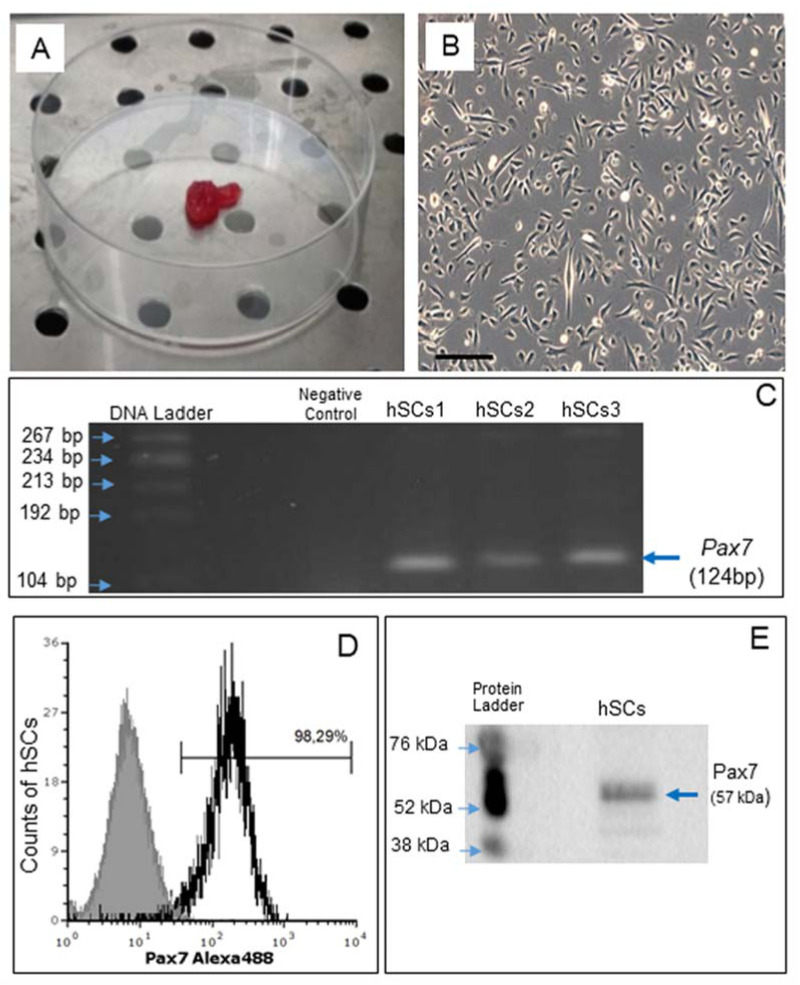
Human skeletal muscle biopsy (**A**) and primary cell culture of hSCs visualized by phase contrast microscopy, original magnification: 10×, scale bar 200 µm (**B**). RT-PCR analysis of *Pax7* gene expression in hSC lines (hSCs1, hSCs2, hSCs3), passage 1. Negative control: PCR without cDNA template (**C**). Representative flow cytometry analysis of Pax7 protein expression in hSC lines, area in black boundaries represents the hSCs stained with primary anti-Pax7 antibody whereas the area under gray shading represents the hSCs only stained with secondary antibody (**D**). Representative Western blot analysis of Pax7, confirming the presence of Pax7 protein in hSCs (**E**).

**Figure 2 ijms-22-07282-f002:**
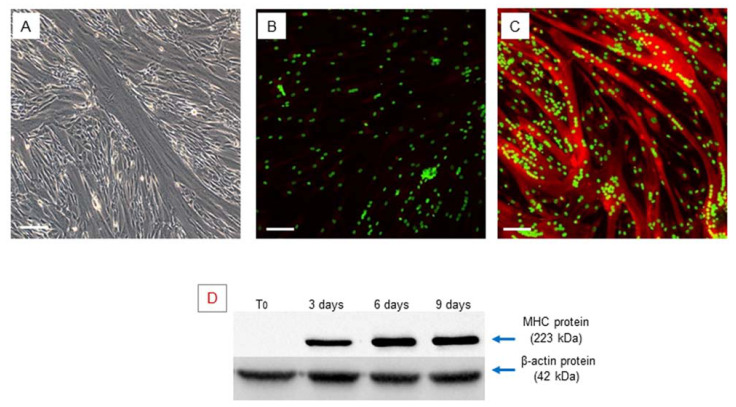
Myogenic differentiation: multinucleated cells (myotubes) observed after 9 days of myogenic induction, visualized by phase contrast microscopy, original magnification: 20×, scale bar 50 µm (**A**). Immunofluorescence staining of MHC in the obtained multinucleated cells. The obtained myotubes showed a high positivity of MHC protein, (**A**) negative control, myotubes stained with secondary antibody only, (**B**) myotubes stained with anti-MHC antibody, LSCM: red for MHC and green for nuclei, original magnification: 10×, scale bar 200 µm. (**B**,**C**). Western blot analysis of MHC during myogenic differentiation. The analysis showed an increase in the *MHC* gene during myogenic differentiation. Experiment was carried out in triplicate and is representative of the three established hSC lines (**D**).

**Figure 3 ijms-22-07282-f003:**
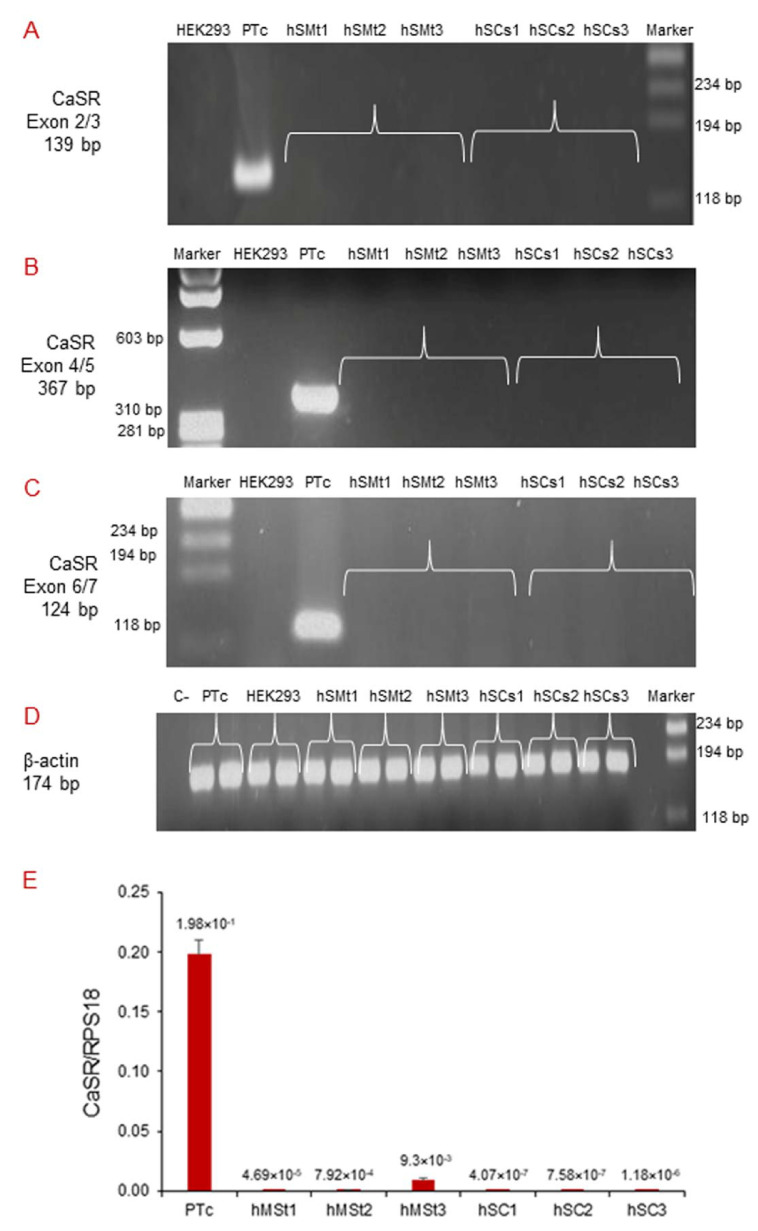
Molecular analysis of *CaSR* gene expression in hSMts and established hSC lines. The analysis in RT-PCR showed the expression of *CaSR* gene using different pairs of primers that fall into: (**A**) exons 2/3; (**B**) exons 4/5; (**C**) exons 6/7. The gene was not detected in any of the samples. The analysis shows the presence of *CaSR* only in human PTcs that were used as a positive control. HEK293 were used as a negative control. Housekeeping *β-actin* gene analysis of the assayed samples (**D**). Real-time qPCR analysis of *CaSR* gene in hSMts and in hSC lines. The analysis showed a very low expression of *CaSR* gene in hSMts and in hSC lines. PTcs were used as positive control (**E**).

**Figure 4 ijms-22-07282-f004:**
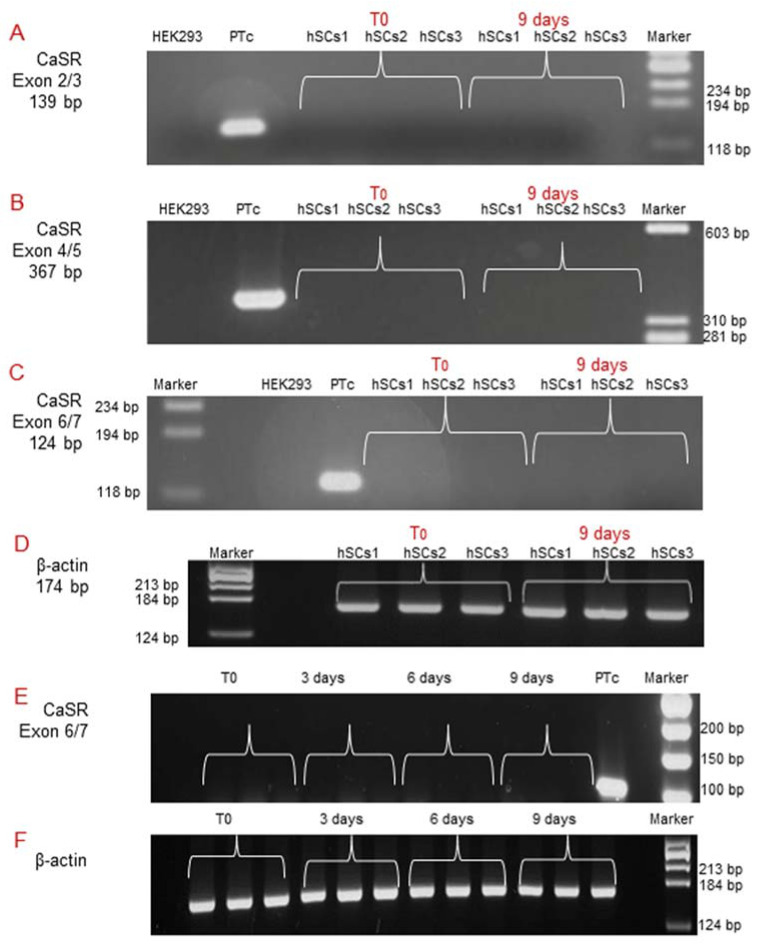
Molecular analysis of *CaSR* gene expression during in vitro myogenesis of the hSC lines. The analysis showed the absence of the *CaSR* gene at T_0_ and at 9 days of myogenic differentiation of 3 primary hSC lines using different pairs of primers that fall into: (**A**) exons 2/3; (**B**) exons 4/5; (**C**) exons 6/7. The primary cultured human PTcs were used as a positive control and HEK293 cells were used as a negative control. Housekeeping *β-actin* gene analysis of the assayed samples (**D**). Real-time qPCR analysis of *CaSR* gene at different time points T_0_, 3, 6, 9 days of myogenesis of the established hSC lines. The analysis showed no detection of expression of *CaSR* gene in hSC lines. Human primary cultured parathyroid cells (PTcs) were used as positive control (**E**). Negative control: PCR without template cDNA in β-actin gene analysis (**F**). Experiments were carried out in triplicate and are representative of the three established hSC lines.

**Figure 5 ijms-22-07282-f005:**
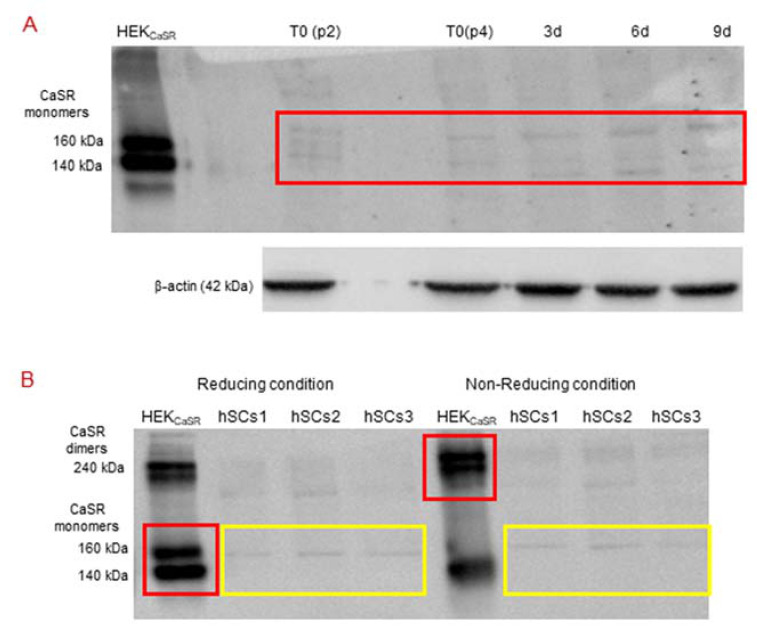
(**A**) Western Blot (standard reducing condition) for CaSR protein in hSC lines and during myogenic differentiation. The results show the presence of faint bands of about 160 kDa and 140 kDa in hSC lines and at different time points (T_0_, 3, 6, 9 days of myogenic differentiation) of primary hSC lines. HEK_CaSR_ were used as positive control. Experiments were carried out in triplicate and are representative of the three established hSC lines. (**B**) Western blot analysis of CaSR in reducing and non-reducing conditions. The analysis for CaSR protein in three hSC lines showed that the observed bands are not CaSR as they did not shift to dimer form at 240 kDa in non-reducing conditions. HEK_CaSR_ were used as positive control. The red and yellow boxes surround the analyzed bands.

**Figure 6 ijms-22-07282-f006:**
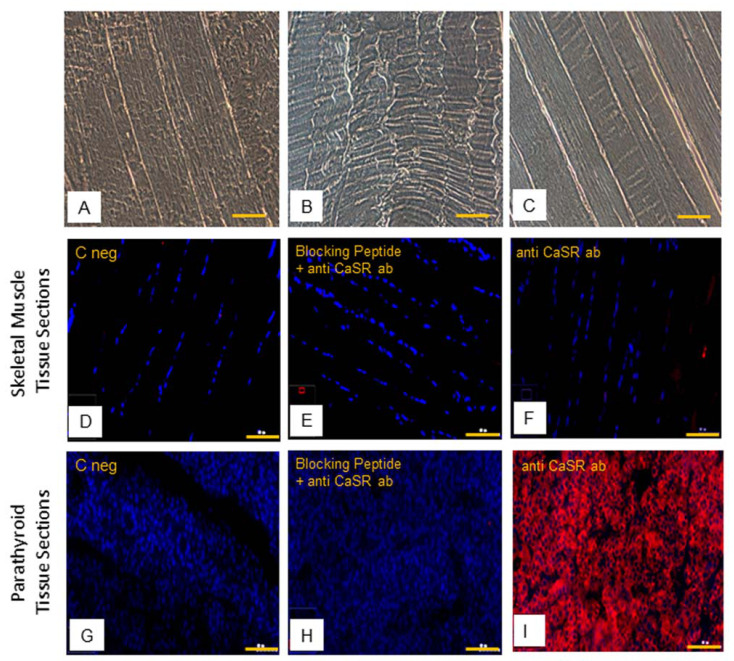
Human skeletal muscle tissue (hSMt) sections; observations in phase contrast microscopy, original. Magnification: 10×, scale bar 200 µm (**A**–**C**). Immunostaining of CaSR protein. Analysis of CaSR protein in human skeletal muscle tissue sections: (**D**) negative control only with secondary antibody, (**E**) control with anti-CaSR antibody absorbed with blocking peptide, and (**F**) with primary anti-CaSR antibody. Analysis of CaSR protein in human parathyroid tissue sections used as positive control: (**G**) negative control with only secondary antibody, (**H**) control with anti-CaSR antibody absorbed with blocking peptide, and (**I**) with primary anti-CaSR antibody. Fluorescent microscopy in conventional colors: red for CaSR and blue for nuclei, original magnification: 10×, scale bar: 200 µm. Experiment was carried out in triplicate (in hSMt sections of three different humans).

**Figure 7 ijms-22-07282-f007:**
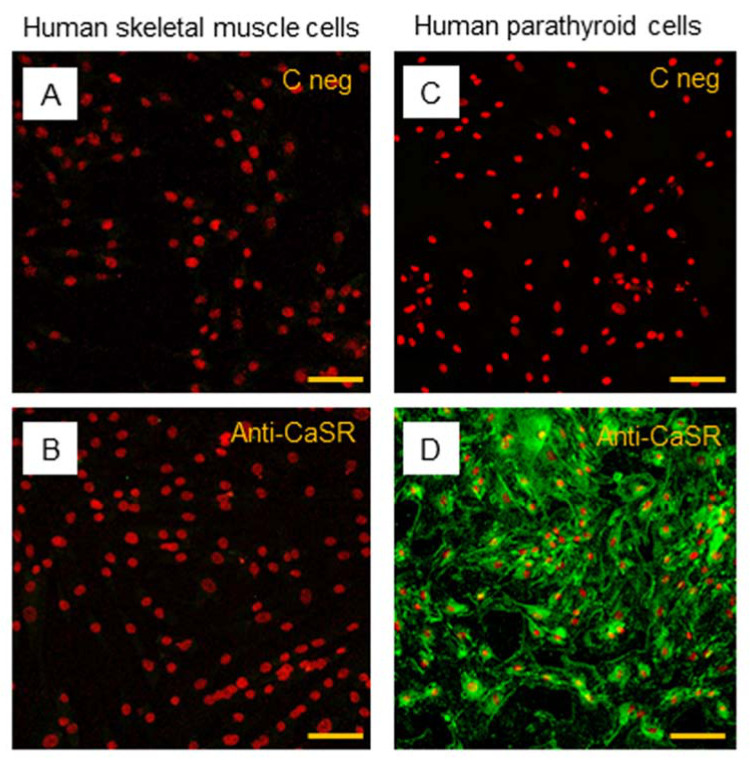
Immunostaining of CaSR in hSC lines. The analysis of CaSR protein in hSC lines: (**A**) negative control with only secondary antibody and (**B**) with primary anti-CaSR antibody. Results are representative of experiments carried out in three established hSC lines. The analysis of CaSR protein was also performed in human parathyroid cells, used as positive control, (**C**) negative control with only secondary antibody, and (**D**) with primary anti-CaSR antibody. LSCM conventional colors: green for CaSR protein and red for nuclei, original magnification: 20×, scale bar: 50 µm.

**Figure 8 ijms-22-07282-f008:**
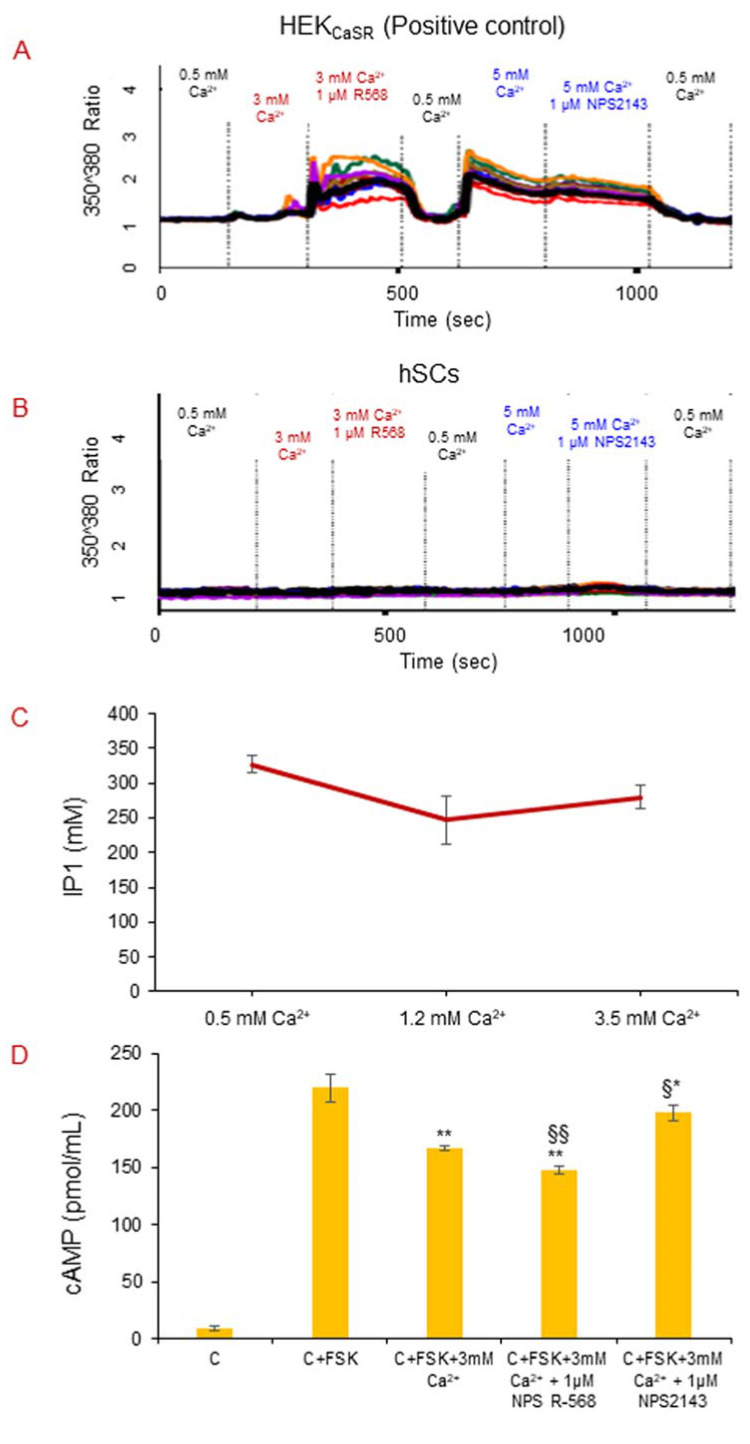
Intracellular calcium mobilization assay with Fura2-AM dye. Graphs representing the effect of CaSR stimulations on intracellular calcium mobilization in HEK_CaSR_ (**A**) used as positive control and hSC lines (**B**). Experiment was carried out in triplicate and is representative of the three established hSC lines. (**C**) IP_1_-Gq ELISA assay in hSC lines. No IP_1_ response was detected in Ca^2+^-induced hSC lines. Experiment was carried out in triplicate and it is representative of the three established hSC lines. (**D**) Gi-cAMP ELISA assay in hSC lines. Graph represents the effect of CaSR stimulation on cAMP response in hSC lines. Experiment was carried out in triplicate and it is representative of three established hSC lines. C: control buffer with 0.5 mM Ca^2+^, FSK: forskolin (positive control), stimulation time: 15 min. * *p* < 0.01 vs. C + FSK; ** *p* < 0.001 vs. C + FSK; §§ *p* < 0.001 vs. C + FSK + 3 mM Ca^2+^; § *p* < 0.005 vs. C + FSK + 3 mM Ca^2+^ (one-way ANOVA with Bonferroni’s post hoc test).

**Figure 9 ijms-22-07282-f009:**
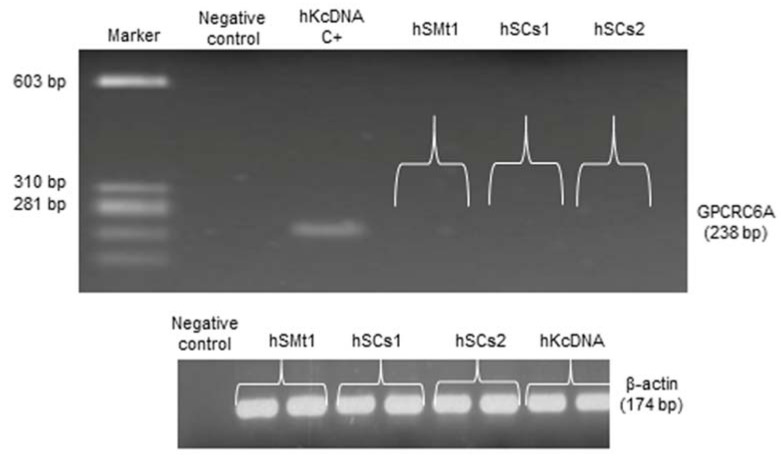
RT-PCR analysis of GPRC6A gene in hSMts and in established hSC lines. Analysis shows the absence of expression of GPRC6A in hSMts and established hSC lines. Commercial human kidney cDNA (hKcDNA) was used as positive control. The negative control was a PCR without templated cDNA.

**Table 1 ijms-22-07282-t001:** RT-PCR analysis of myogenic differentiation marker genes. RT-PCR analysis showed the different expression of *Myf5, MyoD1, Myogenin*, and of *MHC*, analyzed over time during myogenic differentiation of three primary hSC lines (hSCs1, hSCs2, hSCs3).

MRFs Genes	T_0_ HSCs1	T_0_ HSCs2	T_0_ HSCs3	7 Days hSCs1	7 Days hSCs2	7 Days hSCs3
*Myf5*	+	+	+	+	+	+
*MyoD1*	+	+	+	+	+	+
*Myogenin*	--	--	--	+	+	+
*MHC*	--	--	--	+	+	+

**Table 2 ijms-22-07282-t002:** List of the human gene-specific primers used in the analysis.

Name of Gene	Primer Sequence	T_m_ (°C)	Amplicon Size (bp)
*β-actin*	Forward 5′-AGCCTCGCCTTTGCCGA-3′	60 °C	174
	Reverse 5′-CTGGTGCCTGGGGCG-3′		
*Pax7*	Forward 5′-GGTACCGAGAATGATGCGG-3′	55 °C	124
	Reverse 5′-CCCATTGATGAAGACCCCTC-3′		
*Myf5*	Forward 5′-ATGCCATCCGCTACATCG-3′	55 °C	145
	Reverse 5′-ACAGGACTGTTACATTCGGC-3′		
*MyoD1*	Forward 5′-GACGTGCCTTCTGAGTCG-3′	55 °C	148
	Reverse 5′-CTCAGAGCACCTGGTATATCG-3′		
*Myogenin*	Forward 5′-AGCGAATGCAGCTCTCAC-3′	55 °C	150
	Reverse 5′-TGTGATGCTGTCCACGATG-3′		
*MHC*	Forward 5′-GAGTCCTTTGTGAAAGCAACAG-3′	55 °C	143
	Reverse 5′-GCCATGTCCTCGATCTTGTC-3′		
*CaSR exons 2/3*	Forward 5′-GATCAAGATCTCAAATCAAG-3′	57 °C	139
	Reverse 5′-CCAGCGTCAAGTTGGGAAGA-3′		
*CaSR exons 4/5*	Forward 5′-CTGAGAGGTCACGAAGAAAGTG-3′	58 °C	367
	Reverse 5′-GGTGCCAGTTGATGATGGAATA-3′		
*CaSR exons 6/7*	Forward 5′-CTGCTGCTTTGAGTGTGTGG-3′	60 °C	124
	Reverse 5′-CTTGGCAATGCAGGAGGTGT-3′		
*GPRC6A*	Forward 5′-CAGGAGTGTGTTGGCTTTGA-3′	58 °C	238
	Reverse 5′-CTCTTGGCATGTAGCTGGAA-3′		

T_m_: melting temperature; bp: base pairs of amplicon.

**Table 3 ijms-22-07282-t003:** List of the human gene-specific TaqMan probes and primers used in the analysis. TaqMan probes with F as reporter fluorochrome (6-carboxyfuorescein [6-FAM]) and ZEN as quencher. Fluorochrome (Iowa Black FQ).

Name of Gene	Probe and Primer Sequences	Tm (°C)	Amplicon Size (bp)
*RPS18*	Forward 5′-GATGGCAAAGGCTATTTTCCG-3′	58 °C	132
	Reverse 5′-TCTTCCACAGGAGGCCTAC-3′		
	Probe 5′-/56-FAM/TTCAGGGAT/ZEN/CACTAG AGACATGGCTGC/31ABkFQ/-3′		
*CaSR* *exons 6/7*	Forward 5′-TGCTTTGAGTGTGTGGAGTG-3′	58 °C	100
	Reverse 5′-GGTTCTCATTGGACCAGAAGTC-3′		
	Probe 5′-/56-FAM/AGGCACTGG/ZEN/CATCTGTC		
	TCATCAC/31ABkFQ/-3′		

Tm: melting temperature; bp: base pairs of amplicon.

## Data Availability

The data presented in this study are available from the corresponding author upon reasonable request.

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
