# Peer review of "Study of the Expression and Function of Calcium-Sensing Receptor in Human Skeletal Muscle"

_ijms, 2021, doi:10.3390/ijms22147282_

Round 1
Reviewer 1 Report
This is a very interesting paper on the involvement of Calcium-Sensing 22
Receptor (CaSR) in human skeletal muscle development from satellite cells. The rational of the study is clearly sated. I just missed the role of NO, in sarcopenia and its interaction with CaSR.
The study is well designed and the methods are well written and adequate. Results are clearly displayed, figures are informative. The discussion is focused on the results and well written. Overall, this is a well designed and well executed investigation.
Reviewer 2 Report
The manuscript by Romagnoli et al describes the lack of the calcium-sensing receptor in human skeletal muscle. The authors have supported this conclusion by a number of sound methods. This is an important data point for the muscle degeneration inhibiting and regeneration community. To make this point clinically relevant investigations of CaSR expression and function during disease/stress situations must also be conducted.
1) If time permits I would very much like to see expression levels during a pathology, DMD cells, H2O2, TGFb...
2) Line 38. in myopathies it maybe that degeneration is too extensive for regeneration to keep up. In early stages of these diseases regeneration may not be impaired at all, it may just be insufficient.
3) Line 53, "predict in vivo from the in vitro effects"
4) Line 104, "By flow cytometry Pax7 protein.."
5) Figure 2 legend is messed up. A is bright field... also the table explanation is not in the correct place.
6) line 181, sentence has 2 verbs - evaluated and was. Please rewrite
7) Figure 5, this figure would be more convincing if you had loaded a lot less HEK extract. Doing this may even indicate that the bands in question are different sizes.
